# Local correlations necessitate waterfalls as a connection between quasiparticle band and developing Hubbard bands

Juraj Krsnik [1,2] ✉ & Karsten Held [1] ✉

Waterfalls are anomalies in the angle-resolved photoemission spectrum where the energy-momentum dispersion is almost vertical, and the spectrum strongly smeared out. These anomalies are observed at relatively high energies, among others, in superconducting cuprates and nickelates. The prevalent understanding is that they originate from the coupling to some boson, with spin fluctuations and phonons being the usual suspects. Here, we show that waterfalls occur naturally in the process where a Hubbard band develops and splits off from the quasiparticle band. Our results for the Hubbard model with ab initio determined parameters well agree with waterfalls in cuprates and nickelates, providing a natural explanation for these spectral anomalies observed in correlated materials.

Angle-resolved photoemission spectroscopy (ARPES) experiments show, quite universally in various cuprates[1–9], a high energy anomaly in the form of a waterfall-like structure. The onset of these waterfalls is between 100 and 200 meV, at considerably higher energy than the distinctive low-energy kinks[10–12], and they end at even much higher binding energies around ~1 eV[3]. Also, their structure is qualitatively very different: an almost vertical and smeared-out waterfall and not a kink from one linear dispersion to another that is observed at lower binding energies. Akin waterfalls have been reported most recently in nickelate superconductors[13,14], there starting at around 100 meV. This finding puts the research focus once again on this peculiar spectral anomaly. With the close analogy between cuprates and nickelates[15,16] the observation of waterfalls in nickelates gives fresh hope to eventually understand the physical origin of the waterfalls.

Quite similar as for superconductivity, various theories have been suggested for waterfalls in cuprates, including: the coupling to hidden fermions[17], the proximity to quantum critical points[18], and multi-orbital physics[19,20]. The arguably most widespread theoretical understanding is the coupling to a bosonic mode, such as phonons[21] or spin fluctuations (including spin polarons)[22–26]. Here, in contrast to the low energy kinks, the electron-phonon coupling appear a less viable origin for waterfalls, simply because the phonon energy is presumably too low. Also the spin coupling $J$ in cuprates is below 200 meV, which however might concur with the onset of the waterfall. But, its ending at 1 eV is

barely conceivable from a spin fluctuation mechanism, as it is almost an order of magnitude larger than $J$. Even the possibility that waterfalls are matrix element effects that are not present in the actual spectral function has been conjectured[27].

The simplest model for both, superconducting cuprates and nickelates, is the one-band Hubbard model for the Cu(Ni) $3d_{x^2-y^2}$ band. In the case of cuprates, the more fundamental model might be the Emery model which also includes the in-plane oxygen orbitals. However, with some caveats such as doping-depending hopping parameters, a description by the simpler Hubbard model is qualitatively similar[28,29]. In the case of nickelates, these oxygen orbitals are lower in energy, but instead rare earth $5d$ orbitals become relevant and cross the Fermi level[30–34]. Still, the simplest description is that of a one-band Hubbard model plus largely detached $5d$ pockets[35,36]. This simple description is confirmed by ARPES that shows no additional Fermi surfaces and only $5d$ A pockets for $Sr_xLa(Ca)_{1-x}NiO_2$[13,14].

In this paper, we show that waterfalls naturally emerge when a Hubbard band splits off from the central quasiparticle band. This splitting-off is sufficient for, and even necessitates a waterfall-like structure. Using dynamical mean-field theory (DMFT)[37] we can exclude that spin fluctuations are at work, as the feedback of these on the spectrum would require extensions of DMFT[38]. For the doped model, the waterfall prevails in a large range of interactions, which explains its universal occurrence in cuprates and nickelates. A one-on-one

[1]Institute of Solid State Physics, TU Wien, 1040 Vienna, Austria. [2]Department for Research of Materials under Extreme Conditions, Institute of Physics, 10000 Zagreb, Croatia. ✉e-mail: juraj.krsnik@tuwien.ac.at; held@ifp.tuwien.ac.at

comparison of experimental spectra to those of the Hubbard model with ab initio determined parameters for cuprates and nickelates also shows good agreement. Previous papers pointing toward a similar mechanism[9,23,39–45] have, to the best of our knowledge, been quite general, without the more detailed analysis or understanding which the present paper provides. Among others, Macridin et al.[23] noted a positive slope of the DMFT self-energy at intermediate frequencies, but eventually concluded that spin fluctuations lead to waterfalls; Moritz et al.[9,41] emphasized that waterfalls simply connect Hubbard and quasiparticle bands; and Sakai et al.[40] pointed out the importance of the quasiparticle renormalization and vicinity to a Mott transition, advocating the momentum dependence of the self-energy. All these publications use similar numerical quantum Monte Carlo simulations for the Hubbard model either directly for a finite lattice or for lattice extensions of DMFT. In such calculations, it is difficult to track down whether spin fluctuations[23] or other mechanisms[9,40,41] are in charge.

## Results

### Waterfalls in the Hubbard model

Neglecting matrix elements effects, the ARPES spectrum at momentum $\mathbf{k}$ and frequency $\omega$ is given by the imaginary part of Green's function, i.e., the spectral function

$$A(\mathbf{k}, \omega) = -\frac{1}{\pi} \mathrm{Im} \underbrace{\frac{1}{\omega - \varepsilon_{\mathbf{k}} - \Sigma(\omega) + i\delta}}_{\equiv G(\mathbf{k}, \omega)}. \qquad (1)$$

Here, $\delta$ is an infinitesimally small broadening and $\varepsilon_{\mathbf{k}}$ the non-interacting energy-momentum dispersion. For convenience, we set the chemical potential $\mu \equiv 0$. The non-interacting $\varepsilon_{\mathbf{k}}$ is modified by electronic correlations through the real part of the self-energy $\mathrm{Re}\Sigma(\omega)$ while its imaginary part describes a Lorentzian broadening of the poles

(excitations) of Eq. (1) at

$$\omega = \varepsilon_{\mathbf{k}} + \mathrm{Re}\Sigma(\omega). \qquad (2)$$

Please note that we have here omitted the momentum dependence of the self-energy which holds for the DMFT approximation, while non-local correlations can lead to a $\mathbf{k}$-dependent self-energy. This $\mathbf{k}$-dependence can, e.g., arise from spin fluctuations and lead to a pseudogap. In Supplementary Figs. 1 and 2, we compare DMFT to an extension of DMFT, the dynamical vertex approximation (DΓA[46]) that includes such non-local correlations. We show that although non-local correlations can further corroborate the presence of waterfall-like structures, their underlying origin remains tied to local correlations.

Figure 1 shows our DMFT results for the Hubbard model on the two-dimensional square lattice at half-filling with only the nearest neighbor hopping $t$. We go from the weakly correlated regime (left) all the way to the Mott insulator (right). The spectrum then evolves from the weakly broadened and renormalized local density of states (LDOS) resembling the non-interacting system in panel (a) to the Mott insulator with two Hubbard bands at $\pm U/2$ in panel (d). In-between, in panel (c), we have the three-peak structure with both Hubbard bands and a central, strongly-renormalized quasiparticle peak in-between; the hallmark of a strongly correlated electron system that DMFT so successfully describes[37]. Panel (b) is similar to panel (c), with the difference being that the Hubbard bands are not yet so clearly separated. This is the situation where waterfalls emerge in the $\mathbf{k}$-resolved spectrum shown in Fig. 1(j).

### Waterfalls from $\partial \mathrm{Re}\Sigma(\omega)/\partial\omega = 1$

To understand the emergence of this waterfall feature, we solve in Fig. 1(e–h) the pole equation (2) graphically. That is, we plot the right-hand side of Eq. (2), $\varepsilon_{\mathbf{k}} + \mathrm{Re}\Sigma(\omega)$, for three different momenta (colored solid lines), with each momentum indicated by a vertical line of the

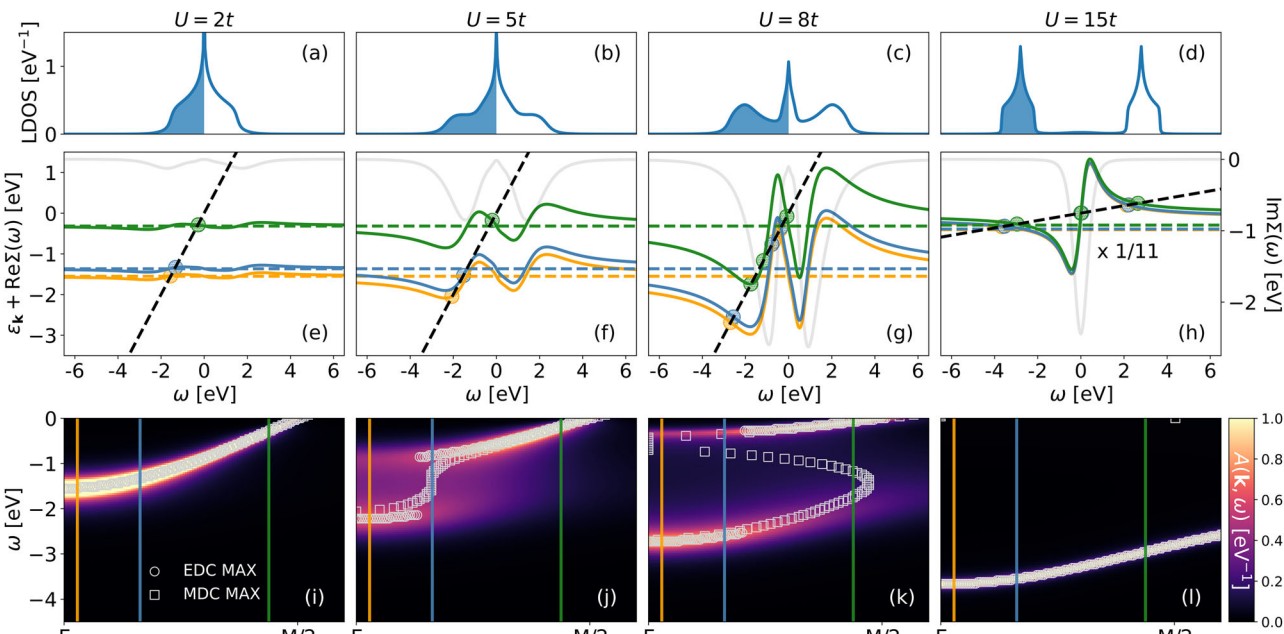

**Fig. 1 | LDOS, graphical solution of the pole equation, and spectral function.** Top (**a–d**): DMFT LDOS for the two-dimensional Hubbard model with nearest neighbor hopping $t = 0.3894$ eV at half-filling and—from left to right—increasing $U$. The shaded areas denote the filled states. Temperature is room temperature $T = t/15$ except for the last column ($U = 15t$) where $T = t$. Energies are in units of eV. Middle (**e–h**): Graphical solution for the poles of the Green's function in Eq. (2) as the crossing point (colored circles) between $\varepsilon_{\mathbf{k}} + \mathrm{Re}\Sigma(\omega)$ (solid lines in three colors for the three $\mathbf{k}$ points indicated by vertical lines in the bottom panel) and $\omega$ (black

dashed line); the colored dashed lines denote $\varepsilon_{\mathbf{k}}$ for the same three momenta. Note that in the insulating case [panel (**h**)] the dashed lines are shifted by $U/2$ and the results scaled by a factor of 1/11. Also shown is the imaginary part of the self-energy (light gray; right $y$-axis). Bottom (**i–l**): $\mathbf{k}$-resolved spectral function $A(\mathbf{k}, \omega)$ along the nodal direction $\Gamma = (0, 0)$ to M $= (\pi, \pi)$, showing a waterfall for $U = 5t$ in panel (**j**). Also plotted are energy distribution curve maxima (EDC MAX, gray circles) and momentum distribution curve maxima (MDC MAX, gray squares) defined as the maxima of $A(\mathbf{k}, \omega)$ as a function of $\omega$ and $\mathbf{k}$, respectively.

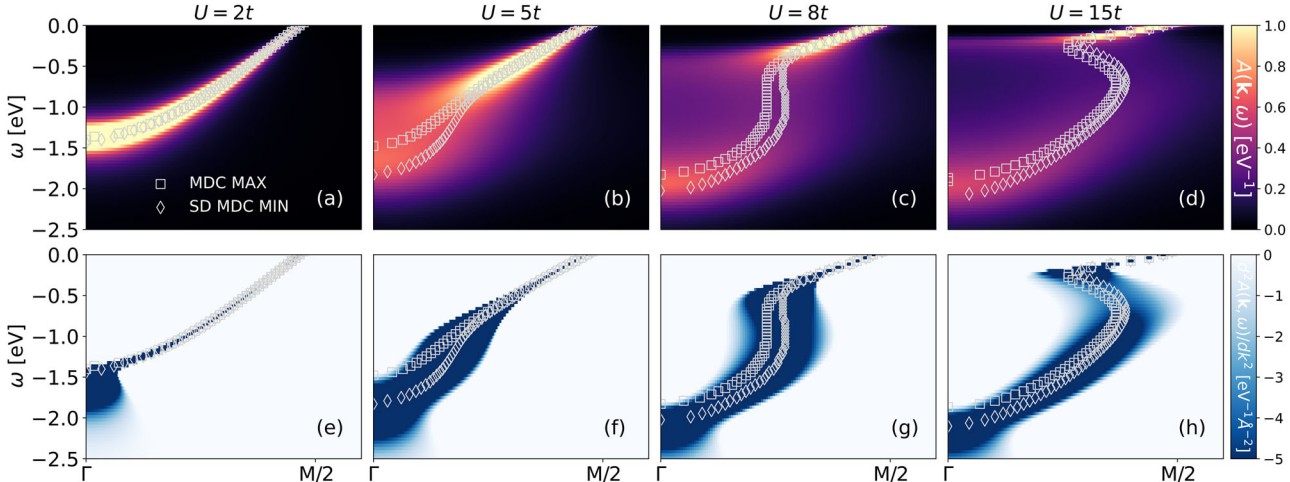

**Fig. 2 | Spectral functions and second derivatives of MDCs for 20% hole doping.** Top (**a**–**d**): Momentum-resolved spectral function for the two-dimensional Hubbard model with nearest neighbor hopping $t = 0.3894$ eV—from left to right—increasing $U$, room temperature $T = t/15$, and 20% hole doping, showing waterfall-like structures in a large interaction range. Bottom (**e**–**h**): Second momentum derivative of the spectral function, which is usually employed in experiments to better visualize the waterfalls. Besides the MDC maxima (MDC MAX, gray squares), we also plot the minima of the second derivative of the MDCs (SD MDC MIN, gray diamonds).

same color in panels (i-l). The left-hand side of Eq. (2), $\omega$, is plotted as a black dashed line. Where they cross, indicated by circles in panels (i-l), we have a pole in the Green's function and a large spectral contribution.

For $U = 2t$ (leftmost column), the excitations are essentially the same as for the non-interacting system $\omega \approx \varepsilon_{\mathbf{k}}$, with the self-energy only leading to a minor quasiparticle renormalization and broadening. In the Mott insulator at large $U$ and zero temperature, on the other hand, $\Sigma(\omega) = U^2/(4\omega)$. Finite $T$ and hopping $t$ regularize this $1/\omega$ pole seen developing in Fig. 1(h), but then turning into a steep positive slope of $\mathrm{Re}\Sigma(\omega)$ around $\omega = 0$. Instead of a delta-function, $\mathrm{Im}\Sigma(\omega)$ becomes a Lorentzian (light gray curve; note the rescaling). Thus, while there is an additional pole-like solution around $\omega = 0$, it is completely smeared out.

Now for $U = 8t$ in Fig. 1(g) we have for large $\omega$ the same pole-like behavior as in the Mott insulator, though of course with a smaller $U^2$ prefactor. On the other hand, at small frequencies $\omega$ we have the additional quasiparticle peak which corresponds to a negative slope $\partial\mathrm{Re}\Sigma(\omega)/\partial\omega|_{\omega=0} < 0$ that directly translates to the quasiparticle renormalization or mass enhancement $m^*/m = 1 - \partial\mathrm{Re}\Sigma(\omega)/\partial\omega|_{\omega=0} > 1$. Altogether, $\varepsilon_{\mathbf{k}} + \mathrm{Re}\Sigma(\omega)$ must hence have the form seen in Fig. 1(g): we have one solution of Eq. (2) at small $\omega$ in the range of the negative, roughly linear $\mathrm{Re}\Sigma(\omega)$, which corresponds to the quasiparticle excitations. We have a second solution at large $\omega$, where we have the $1/\omega$ self-energy as in the Mott insulator, which corresponds to the Hubbard bands. For a chosen $\mathbf{k}$, there is a third crossing in-between, where the self-energy crosses from the Mott like $1/\omega$ to the quasiparticle like $-\omega$ behavior. Here, the self-energy has a positive slope. This pole is however not visible in $A(\mathbf{k}, \omega)$ [Fig. 1(k)], simply because the smearing $\mathrm{Im}\Sigma(\omega)$ is very large. It would not be possible to see it in ARPES.

However, numerically, one can trace it as the maximum in the momentum distribution curve (MDC), i.e, $\max_{\mathbf{k}}A(\mathbf{k}, \omega)$ along $\Gamma$ to M, shown as squares in Fig. 1k. This MDC shows an S-like shape since the positive slope of $\mathrm{Re}\Sigma(\omega)$ in this intermediate $\omega$ range is larger than one (dashed black line). Consequently, for $\varepsilon_{\mathbf{k}}$ at the bottom of the band (orange and blue lines) this third pole in panel (g) is close to the quasiparticle pole, while for $\varepsilon_{\mathbf{k}}$ closer to the Fermi level $\mu \equiv 0$ (green line) it is close to the pole corresponding to the Hubbard band.

For the smaller $U$ of Fig. 1(e), on the other hand, $\Sigma$ is small and thus also the positive slope in the intermediate $\omega$ range must be smaller than one (dashed black line). Together with the continuous evolution of the self-energy from (e) to (j), this necessitates that for some Coulomb interaction in-between, the slope close to the inflection point in-between Hubbard and quasiparticle band equals one: $\partial\mathrm{Re}\Sigma(\omega)/\partial\omega = 1$. That is the case for $U \approx 5t$ shown in Fig. 1(k).

Now there is only one pole for each momentum. For the momentum closest to the Fermi level $\mu$ (green line), it is in the quasiparticle band where $\mathrm{Re}\Sigma(\omega) \sim -\omega$ at small $\omega$. When we reduce $\varepsilon_{\mathbf{k}}$, i.e., shift the $\varepsilon_{\mathbf{k}} + \mathrm{Re}\Sigma(\omega)$ curve down, there is one momentum (blue curve) where the crossing is not in the quasiparticle band nor in the Hubbard band but in the crossover region between the two, with the positive slope of $\mathrm{Re}\Sigma(\omega)$. As this slope is one, the blue and black dashed lines are close to each other in a large energy region. That is, we are close to a pole for many different energies $\omega$. Given the finite imaginary part of the self-energy, we are thus within reach of an actual pole. Consequently, we get a waterfall in Fig. 1(j) with spectral weight in a large energy range for this blue momentum. Finally, for $\varepsilon_{\mathbf{k}}$'s at the bottom of the band (orange line), the crossing point is in the lower Hubbard band. Altogether this leads to a waterfall as a crossover from the quasiparticle to the Hubbard band.

**Doped Hubbard model**

Next, we turn to the doped Hubbard model in Fig. 2. The main difference is that now for the $U \to \infty$ limit, we do not get a Mott insulator, but keep a strongly correlated metal. As a consequence the $U$-range where we have waterfall-like structures is much wider, which explains that they are quite universally observed in cuprates and nickelates. Strictly speaking, an ideal vertical waterfall again corresponds mathematically to a slope one close to the inflection point of $\mathrm{Re}\Sigma(\omega)$. This is the case for $U \approx 8t$ in Fig. 2c. However, with the much slower evolution with $U$ at finite doping, we have a large $U$ range with waterfall-like structures, first at small $U$ in the form of moderate slopes as in panel (b), and then for large $U$ in form of an S-shape-like structure as in panel (d) that are akin to waterfalls. The survival of the S-shape structure even at $U = 15t$ strongly suggests that it extends up to $U \to \infty$.

Figure 2 (e-h) shows the second derivative (SD) of the MDC, i.e., $\partial^2 A(\mathbf{k}, \omega)/\partial\mathbf{k}^2$ along the momentum line $\Gamma$ to M. This SD MDC is usually used in an experiment to better visualize the waterfalls; and indeed we see in Fig. 2e–h that the waterfall becomes much more pronounced and better visible than in the spectral function itself.

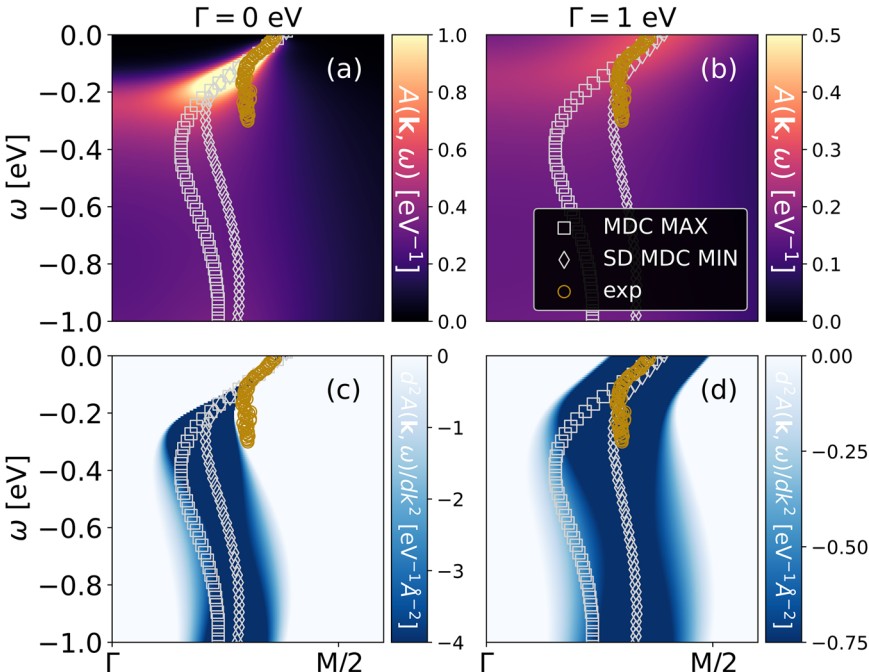

**Fig. 3 | DFT+DMFT calculations of waterfalls in nickelates.** Waterfalls in the one-band DMFT spectrum (**a**; top) and its second derivative (**c**; bottom) for $Sr_{0.2}La_{0.8}NiO_2$, compared to experiment[13] (exp, golden circles), together with the MDC maxima (MDC MAX, gray squares), and the minima of the second derivative of the MDCs (SD MDC MIN, gray diamonds). In the right column (**b**; **d**), we added a broadening $\Gamma = 1$ eV to the DMFT self-energy to mimic disorder effects. The ab initio determined parameters of the Hubbard model for nickelates are[35]: $t = 0.3894$ eV, $t' = -0.25t$, $t'' = 0.12$, $U = 8t$, 20% hole doping, and we take a sufficiently low temperature $T = 100/t$.

## Connection to nickelates and cuprates

Let us finally compare our theory for waterfalls to ARPES experiments for nickelates and cuprates. We here refrain from adjusting any parameters and use the hopping parameters of the Hubbard model that have been determined before ab initio by density functional theory (DFT) for a one-band Hubbard model description of $Sr_{0.2}La_{0.8}NiO_2$[35], $La_{2-x}Sr_xCuO_4$[47], and $Bi_2Sr_2CuO_6$ (Bi2201)[48]. Similarly, constrained random phase approximation (cRPA) results are taken for the interaction $U$[34,48]. CRPA $U$ values of ref. 47 are rounded up similarly to nickelates as in refs. 34,35 to mimic the frequency dependence of $U$. The parameters are listed in the captions of Figs. 3 and 4.

Figure 3 compares the waterfall structure in the one-band Hubbard model for nickelates to the ARPES experiment[13] (for waterfalls in nickelates under pressure cf.[45]). The qualitative agreement is very good. Quantitatively, the quasiparticle renormalization is also well described without free parameters. The onset of the waterfall is at a similar binding energy as in ARPES, though a bit higher, and at a momentum closer to $\Gamma$.

This might be due to different factors. One is that nickelate films still have a high degree of disorder, especially stacking faults. We can emulate this disorder by adding a scattering rate $\Gamma$ to the imaginary part of the self-energy. For $\Gamma = 1$ eV, we obtain Fig. 3b, d which is on top of experiment also for the waterfall-like part of spectrum, though with an adjusted $\Gamma$. Indeed, we think that this $\Gamma$ is a bit too large, but certainly disorder is one factor that shifts the onset of the waterfall to lower binding energies. Other possible factors are (i) the $\omega$-dependence of $U(\omega)$ in cRPA which we neglect, and (ii) surface effects on the experimental side to which ARPES is sensitive. Also (iii) a larger $U$ would according to Fig. 2 result in an earlier onset of the waterfall. At the same time, it would however also increase the quasiparticle renormalization which is, for the predetermined $U$, in good agreement with the experiment.

Figure 4 compares the DMFT spectra of the Hubbard model to the energy-momentum dispersions extracted by ARPES for two cuprates. Panels (a-d;g-j) show the comparison for four different dopings $x$ of $La_{2-x}Sr_xCuO_4$. Again, we have a good qualitative agreement including

the change of the waterfall from a kink-like structure at large doping $x = 0.3$ in panels (d,j) to a more S-like shape at smaller doping $x = 0.12$ in panels (a,g). The same doping dependence is also observed for Bi2201 from panels (f,l) to (e,k). Note, lower doping effectively means stronger correlations, similar to increasing $U$ in Fig. 2, where we observe the same qualitative change of the waterfall. Altogether this demonstrates that even changes in the form of the waterfall from kink-like to vertical waterfalls to S-like shape can be explained. Quantitatively, we obtain a very good agreement at larger dopings, while at lower dopings there are some quantitative differences. However, please keep in mind that we did not fit any parameters here.

## Discussion

### Umbilical cord metaphor

At small interactions $U$ all excitations or poles of the Green's function are within the quasiparticle band; for very large $U$ and half-filling all are in the Hubbard bands; and for large, but somewhat smaller $U$ we have separated quasiparticle and Hubbard bands. We have proven that there is a qualitatively distinct fourth "*waterfall*" parameter regime. Here, the Hubbard band is not yet fully split off from the quasiparticle band, and we have a crossover in the spectrum from the Hubbard to the quasiparticle band in the form of a waterfall. This waterfall must occur when turning on the interaction $U$ and is, in the spirit of Ockham, a simple explanation of the waterfalls observed in cuprates, nickelates, and other transition metal oxides. Even the change from a kink-like to an actual vertical waterfall to an S-like shape with increasing correlations agrees with the experiment.

As Supplementary Movie 1, we provide a movie of the spectrum evolution with increasing $U$. Figuratively, we can call this evolution the "*birth of the Hubbard band*", with the quasiparticle band being the "*mother band*" and the Hubbard band the "*child band*". The waterfall is then the "*umbilical cord*" connecting the "*mother band*" and "*child band*" before the latter becomes fully disconnected from the former. As a matter of course, such metaphors are never perfect. Here, e.g., we rely on the time axis being identified with increasing $U$. However, one

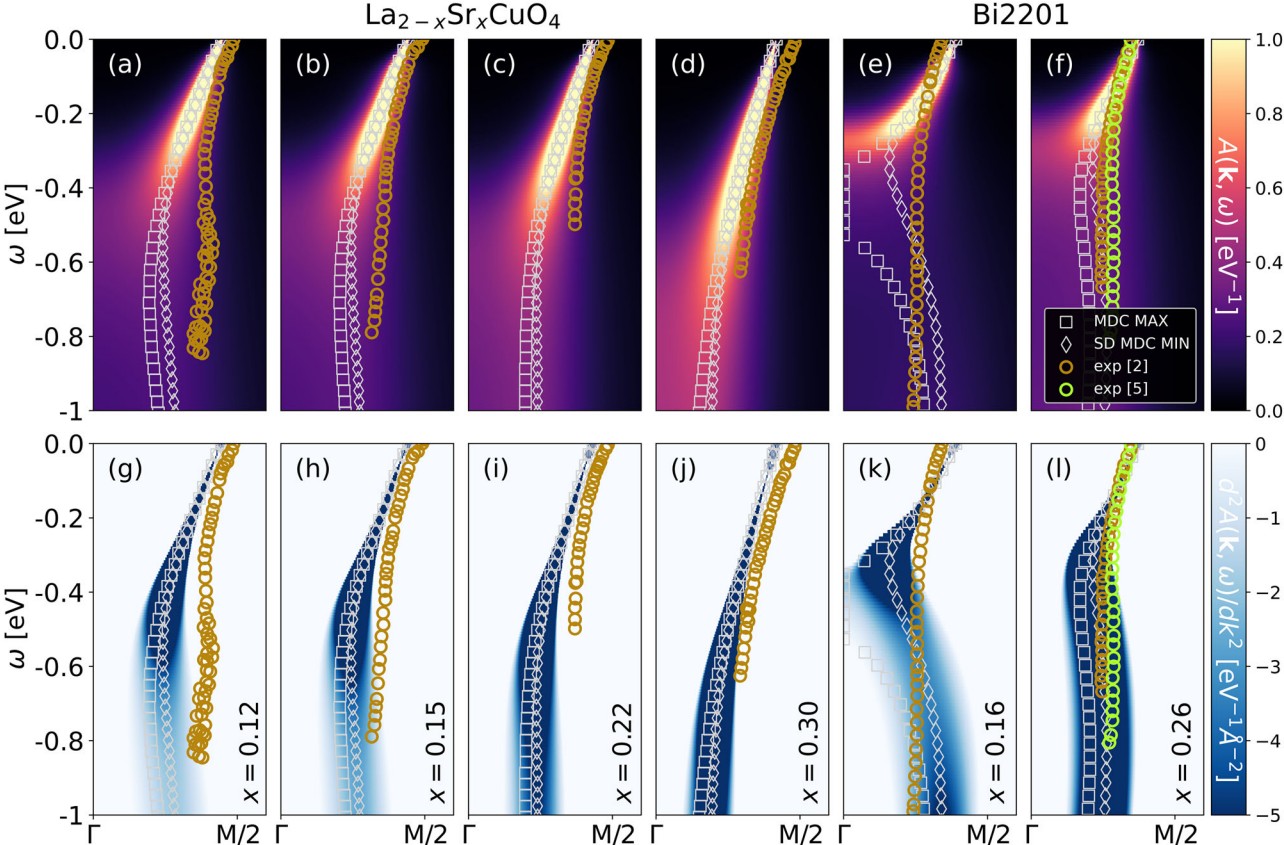

**Fig. 4 | DFT+DMFT calculations of waterfalls in cuprates.** Waterfalls in the DMFT spectrum (top) and its second derivative (bottom) for (**a–d;g–j**) La$_{2-x}$Sr$_x$CuO$_4$ at four different $x$ and (**e, f;k, l**) Bi$_2$Sr$_2$CuO$_6$ (Bi2201) for $x = 0.16$ and $x = 0.26$ hole doping compared to experiment (golden circles, ref. 2), (green circles, ref. 5). Also shown are the MDC maxima (MDC MAX, gray squares), and the minima of the second derivative of the MDCs (SD MDC MIN, gray diamonds). The ab initio determined parameters of the one-band Hubbard model for La$_{2-x}$Sr$_x$CuO$_4$ are[47]: $t = 0.4437$ eV, $t' = -0.0915t$, $t'' = 0.0467t$, $U = 7t$. CRPA $U$ values of ref. 47 are rounded up similarly to nickelates as in refs. 34,35 to mimic the frequency dependence of $U$. Those for Bi2201 are[48]: $t = 0.527$ eV, $t' = -0.27t$, $t'' = 0.08t$, $U = 8t$ (rounded). Again we set $T = 100/t$.

could also interpret it vice versa, that is, as the quasiparticle band disconnects from the Hubbard band as $U$ decreases.

## Methods

In this section, we outline the model and computational methods employed. The two-dimensional Hubbard model for the $3d_{x^2-y^2}$ band reads

$$\mathcal{H} = \sum_{ij\sigma} t_{ij} \hat{c}_{i\sigma}^\dagger \hat{c}_{j\sigma} + U \sum_i \hat{n}_{i\uparrow} \hat{n}_{i\downarrow}. \qquad (3)$$

Here, $t_{ij}$ denotes the hopping amplitude from site $j$ to site $i$, which we restrict to nearest neighbor $t$, next-nearest neighbor $t'$, and next-next-nearest neighbor hopping $t''$; $\hat{c}_i^\dagger$ ($\hat{c}_j$) are fermionic creation (annihilation) operators, and $\sigma$ marks the spin; $\hat{n}_{i\sigma} = \hat{c}_{i\sigma}^\dagger \hat{c}_{i\sigma}$ are occupation number operators; $U$ is the Coulomb interaction.

DMFT calculations were done using `w2dynamics`[49] which uses quantum Monte Carlo simulations in the hybridization expansion[50]. For the analytical continuation, we employ maximum entropy with the chi2kink method as implemented in the `ana_cont` code[51].

## Data availability

The data that support the findings of this study are available in with the identifier(s)[52]. This also includes some digitized experimental data points from refs. 2,5,13,14.

## Code availability

The `w2dynamics` code[49] is available at github.com/w2dynamics/w2dynamics; the `ana_cont` code[51] is available at github.com/josefkaufmann/ana_cont.

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

## Acknowledgements

The authors thank Simone Di Cataldo, Andreas Hausoel, Eric Jacob, Oleg Janson, Motoharu Kitatani, Liang Si, Paul Worm, and Yi-feng Yang for helpful discussions. KH and JK acknowledge funding by the Austrian Science Funds (FWF) through Grant DOI 10.55776/P36213. KH further acknowledges funding through FWF Grant DOI 10.55776/I5398, SFB Q-M&S (FWF Grant DOI 10.55776/F86), and Research Unit QUAST by the Deutsche Foschungsgemeinschaft (DFG project ID FOR5249; FWF Grant DOI 10.55776/I5868). The DMFT calculations have been done in part on the Vienna Scientific Cluster (VSC).

## Author contributions

J.K. did the DMFT calculations, analytic continuations, and designed the figures; K.H. devised and supervised the project, and did a major part of the writing. Both authors discussed and refined the project, arrived at the physical understanding presented, and approved the submitted version.

## Competing interests

The authors declare no competing interests.
