## [Transparent Peer Review file · Nature Communications]

Local correlations necessitate waterfalls as a connection between quasiparticle band and developing Hubbard bands

Corresponding Author: Professor Karsten Held

Version 0:

Reviewer comments:

Reviewer #2

(Remarks to the Author)

The paper by Krsnik and Held scrutinizes a so-called waterfall behavior seen in the two-dimensional Hubbard model in relation to those observed in the angle-resolved photoemission spectra of cuprates and nickelates. They use the dynamical mean-field theory (DMFT) to calculate the self-energy and spectral function for various values of U and doping concentration, and find that the waterfall behavior appears in the intermediate regime between the weakly correlated region and the strongly correlated region where the Hubbard bands well develop. The comparisons with the experimental results on cuprates and nickelates show qualitative and semi-quantitative consistency.

The paper is well-written and scientifically sound. The results look consistent with the previous works using similar numerical methods. The virtue of this paper is the detailed study of this previously found behavior, clarifying the mechanism and showing the universality of the phenomena, as well as the consistency with the experimental observations. Because the work contributes to clarifying the mechanism of the waterfall behavior, I think the paper is worth publication in Nature Communications though I have several comments as described below.

1. In Eq.(1), the authors neglect the momentum dependence of the self-energy. This is only true under the approximation of the DMFT, which is not mentioned around Eq.(1). I suggest the authors clarify the general formula and approximation they take.
2. In relation to the comment 1, it is in general important to take account of the momentum dependence of the self-energy in the two-dimensional system. In particular, in this paper, they discuss the change of the dispersion due to the self-energy, where it will be reasonable to expect the effect of the momentum-dependent self-energy. Will the main conclusion in this paper be unchanged by taking account of the momentum dependence of the self-energy?
3. In relation to the above comments, there are several preceding studies on the waterfall behavior with cluster extensions of the DMFT, where the spatial correlations beyond the DMFT are taken into account: Figure 2b in Ref.23 looks pointing out a mechanism similar to that suggested by the authors of the present manuscript, though a detailed discussion is lacking. Figure 6 in PRB 82, 134505 (2010) compares the theory with the experimental results on cuprates and shows a semi-quantitative consistency, though a detailed discussion on the mechanism is lacking. I think the authors should add more precise descriptions on the previous works and discriminate their works from them.
4. In Fig.1 the authors identify the self-energy structure producing the waterfall behavior, and discuss its relevance to real materials. Although the realization of the self-energy structure in Fig.1(b) seems to require a special strength of the correlations, they argue in Sec.III that a structure close to Fig.1(b) can universally occur because of a much slower change with U at finite doping. Considering the point 3 above, I think it is important to show this argument more quantitatively. For instance, it will be interesting if the authors could plot the U range of the robust waterfall behavior against doping.
5. In Sec.IV, the authors discuss the origin of the quantitative discrepancy between theory and experiments seen in Figs.3 and 4. I wonder if my comments 1 and 2 may also be relevant to this discrepancy. The lack of spatial correlations could also affect the appropriate choice of the U value, which could be larger than the cRPA one.

Reviewer #3

(Remarks to the Author)

Report for the manuscript "Waterfalls: Umbilical Cords at the Birth of Hubbard Bands" submitted to Nature Communications by Juraj Kršnik and Karsten Held

The manuscript investigates spectroscopic anomalies, known as waterfalls, observed in ARPES experiments of superconducting cuprates and nickelates. These anomalies are characterized by nearly vertical energy-momentum dispersion and a smeared-out spectral appearance, occurring at energies between 100 to 200 meV and extending up to around 1 eV.

The authors primarily utilize the single-band Hubbard model, analyzed through single site dynamical mean-field theory, to explain the emergence of these waterfall features. They demonstrate that waterfalls naturally occur when a Hubbard band splits off from the quasiparticle band — a process driven by the local electronic correlations captured by the DMFT self-energy. The authors argue that this band-splitting process is sufficient to explain the waterfall phenomenon, without the need to invoke more complex mechanisms, such as coupling to bosonic modes like phonons or spin fluctuations. Additionally, the manuscript references and compares its findings to several key studies on cuprates and nickelates, showing that the Hubbard model's predictions align well with experimental observations. The authors emphasize that the observation of waterfalls within the DMFT framework—despite its exclusion of non-local spin excitations—suggests that these features can arise purely from local correlations. This provides a simplified explanation for the waterfalls, which the authors present as an inherent feature of the Hubbard model.

The main results of the paper are clear and significant. The authors convincingly demonstrate that the waterfall features observed in ARPES experiments can be explained by the formation and splitting of a Hubbard band from the quasiparticle band within the single-band Hubbard model. This finding is important because it simplifies the understanding of the spectral anomalies by attributing them to local electronic correlations, as captured by the self-energy in DMFT. An interesting aspect of this work is the implicit use of DMFT's limitations as a strength. DMFT, by design, includes only local electronic correlations and does not account for non-local spin excitations in the self-energy. The authors capitalize on this limitation by showing that the waterfall phenomena can still be captured within this framework. This implies that the occurrence of waterfalls does not necessarily require the coupling of quasiparticles to non-local spin excitations, such as magnons. In a sense, the authors have turned a shortcoming of the DMFT approximation into a compelling feature of their analysis, reinforcing their conclusions.

I can wholeheartedly recommend the publication of this manuscript in Nature Communications due to its clear, simple yet broad key message, and its potential to make an important impact on a wide range of research in correlated electron systems, both experimental and theoretical. The authors present a compelling case that the waterfall features observed in ARPES spectra can be understood without invoking additional bosonic degrees of freedom. This insight is valuable and could influence future studies in the field.

However, I do have reservations regarding the use of the anthropomorphized metaphor of the "umbilical cord," and the title's reference to the "birth" of Hubbard bands. While such creative metaphors can be effective in conveying complex ideas, I find that this particular metaphor does not resonate well with the underlying physics. Specifically, the suggestion that Hubbard bands are "born" out of a non-interacting reference system seems artificial to me. In fact, one might argue the opposite: Hubbard bands are closely related to atomic-like excitations, while the low-energy quasiparticles are the emergent phenomenon when individual atoms are assembled into a solid. That said, this is a non-scientific critique, and I do not wish to dwell on it. The bottom line is that - in my opinion - the metaphor employed does not carry sufficient depth or accuracy to justify its use. I would suggest the authors reconsider the metaphor and perhaps choose a more fitting title that better reflects the scientific content and significance of their findings.

Version 1:

Reviewer comments:

Reviewer #2

(Remarks to the Author)

The authors replied to my previous comments satisfactorily and revised the manuscript appropriately. I now recommend a publication of the manuscript.

Reviewer #3

(Remarks to the Author)

Second report for the manuscript "Waterfalls: Umbilical Cords at the Birth of Hubbard Bands" submitted to Nature Communications by Juraj Kršnik and Karsten Held

The authors replied to my previous comment in a satisfactory way. While I still fail to see a great depth in the metaphor, this issue is more a matter of personal taste and should not affect the conclusion.

I now recommend publication of the manuscript as it is.

REPLY TO REVIEWER #2

Reviewer #2: The paper by Krsnik and Held scrutinizes a so-called waterfall behavior seen in the two-dimensional Hubbard model in relation to those observed in the angle-resolved photoemission spectra of cuprates and nickelates. They use the dynamical mean-field theory (DMFT) to calculate the self-energy and spectral function for various values of U and doping concentration, and find that the waterfall behavior appears in the intermediate regime between the weakly correlated region and the strongly correlated region where the Hubbard bands well develop. The comparisons with the experimental results on cuprates and nickelates show qualitative and semi-quantitative consistency.

The paper is well-written and scientifically sound. The results look consistent with the previous works using similar numerical methods. The virtue of this paper is the detailed study of this previously found behavior, clarifying the mechanism and showing the universality of the phenomena, as well as the consistency with the experimental observations. Because the work contributes to clarifying the mechanism of the waterfall behavior, I think the paper is worth publication in Nature Communications though I have several comments as described below.

We sincerely thank the Reviewer for their very positive assessment of our work. In the following, we are pleased to reply to all of the Reviewer's comments. We believe that additional discussions brought by the Referee further elevate the significance of our work and help to differentiate it from prior results.

Reviewer #2: 1.: In Eq.(1), the authors neglect the momentum dependence of the self-energy. This is only true under the approximation of the DMFT, which is not mentioned around Eq.(1). I suggest the authors clarify the general formula and approximation they take.

That is a good point of the Reviewer. As suggested, we now clarify that in general the self-energy is a momentum-dependent quantity, while DMFT approximates it as momentum-independent.

Reviewer #2: 2.: In relation to the comment 1, it is in general important to take account of the momentum dependence of the self-energy in the two-dimensional system. In particular, in this paper, they discuss the change of the dispersion due to the self-energy, where it will be reasonable to expect the effect of the momentum-dependent self-energy. Will the main conclusion in this paper be unchanged by taking account of the momentum dependence of the self-energy?

This is a good point by the Reviewer. In general, given the high energy of the waterfalls, and the small energy of non-local correlation effects such as spin fluctuations, we expect deviations from DMFT in the waterfall energy region to be relatively small.

To address this point in a more thorough way, we have done additional calculations using the dynamical vertex approximation (D Γ A), which is an extension of DMFT that accounts for non-local correlations. The results are presented in the Supplemental Information. Within D Γ A, we also observe the presence of waterfalls in the spectra, and importantly, find that their origins can be traced back to local correlations already captured by DMFT. In particular, even if a gap opens at low energies due to strong spin fluctuations, we find a qualitative and semi-quantitative agreement with DMFT regarding waterfalls.

Reviewer #2: 3.: In relation to the above comments, there are several preceding studies on the waterfall behavior with cluster extensions of the DMFT, where the spatial correlations beyond the DMFT are taken into account: Figure 2b in Ref.23 looks pointing out a mechanism similar to that suggested by the authors of the present manuscript, though a detailed discussion is lacking. Figure 6 in PRB 82, 134505 (2010) compares the theory with the experimental results on cuprates and shows a semi-quantitative consistency, though a detailed discussion on the mechanism is lacking. I

think the authors should add more precise descriptions on the previous works and discriminate their works from them.

We thank the Reviewer for drawing our attention to PRB 82, 134505 (2010) which we now cite in our manuscript and to Fig. 2b of Ref. 23. As suggested by the Reviewer, we extended our discussion on the previous works, i.e., that the proximity to the Mott transition, the connection between Hubbard and quasiparticle band and the self-energy slope have been connected before with waterfalls.

However, as the Reviewer writes, a detailed understanding was still lacking, leading to the observation of waterfalls, but their interpretations only vaguely discussed, and predominantly associated with non-local correlations. In particular, in PRB 82, 134505 (2010) the vicinity to a Mott transition was recognized, however, also noted that “This unusual structure comes out thanks to the use of the cumulant periodization see Sec. II which can describe the strong momentum dependence of the self-energy”. Moreover, in Fig. 2b of Ref. 23, the slope of $\text{Re}\Sigma$ within DMFT was noted to be roughly 1 for the spectrum with a waterfall, but this mechanism was abandoned in favor of spin fluctuations due to the good agreement of renormalized second-order approximation with Monte Carlo results. Let us note that Green’s function used in the second-order self-energy was already interacting, as we understand the calculation, presumably containing from the outset waterfall features. Adding on top spin fluctuations could then enhance the waterfall effects, similar to our DGA results in Supplementary Information. Nevertheless, as the Reviewer mentions “a detailed discussion” such as ours is “lacking”.

Reviewer #2: 4.: In Fig.1 the authors identify the self-energy structure producing the waterfall behavior, and discuss its relevance to real materials. Although the realization of the self-energy structure in Fig.1(b) seems to require a special strength of the correlations, they argue in Sec.III that a structure close to Fig.1(b) can universally occur because of a much slower change with U at finite doping. Considering the point 3 above, I think it is important to show this argument more quantitatively. For instance, it will be interesting if the authors could plot the U range of the robust waterfall behavior against doping.

We agree with the Reviewer that quantifying this statement is helpful. We have extended the calculations for 20% hole doping to very large Coulomb interaction $U = 15$, which we now show instead of the $U = 10t$ case in the main text. This evidences that the waterfalls (of the “S”-like shape) survive in the large U limit, at least for all practical purposes. Drawing a phase diagram is difficult since the waterfall sets in and disappears smoothly, and we could not come up with a clear-cut mathematical criterion for a “robust waterfall”.

Reviewer #2: 5.: In Sec.IV, the authors discuss the origin of the quantitative discrepancy between theory and experiments seen in Figs.3 and 4. I wonder if my comments 1 and 2 may also be relevant to this discrepancy. The lack of spatial correlations could also affect the appropriate choice of the U value, which could be larger than the cRPA one.

Both, non-local correlations as well as an underestimation of the U blue, might be responsible for the only-semi qualitative agreement. Further, the Hubbard model might not be fully appropriate for cuprates, nickelates are still suffering from huge amounts of disorder, and there are also uncertainties in the determined t and t' parameters.

Please keep in mind that we did not at all try to adjust any parameters for better agreement with experiments. Given this conservative approach, the semi-quantitative agreement we get is quite convincing in our view; all one could have asked for. Whether the remaining difference is due to differences in U , non-local correlations (as suggested by the Reviewer), different hopping parameters including a finite hopping in the z direction, disorder, or using the more appropriate Emery model instead of the Hubbard model, is difficult to say.

We feel that it is too difficult to trace down the precise origin for the remaining disagreement with experiment, and just listing the above possibilities, maybe even overlooking other possible origins, does not really serve the reader.

I. REPLY TO REVIEWER #3

Reviewer #3: Report for the manuscript "Waterfalls: Umbilical Cords at the Birth of Hubbard Bands" submitted to Nature Communications by Juraj Kršnik and Karsten Held

The manuscript investigates spectroscopic anomalies, known as waterfalls, observed in ARPES experiments of superconducting cuprates and nickelates. These anomalies are characterized by nearly vertical energy-momentum dispersion and a smeared-out spectral appearance, occurring at energies between 100 to 200 meV and extending up to around 1 eV.

The authors primarily utilize the single-band Hubbard model, analyzed through single site dynamical mean-field theory, to explain the emergence of these waterfall features. They demonstrate that waterfalls naturally occur when a Hubbard band splits off from the quasiparticle band — a process driven by the local electronic correlations captured by the DMFT self-energy. The authors argue that this band-splitting process is sufficient to explain the waterfall phenomenon, without the need to invoke more complex mechanisms, such as coupling to bosonic modes like phonons or spin fluctuations. Additionally, the manuscript references and compares its findings to several key studies on cuprates and nickelates, showing that the Hubbard model's predictions align well with experimental observations. The authors emphasize that the observation of waterfalls within the DMFT framework—despite its exclusion of non-local spin excitations—suggests that these features can arise purely from local correlations. This provides a simplified explanation for the waterfalls, which the authors present as an inherent feature of the Hubbard model.

The main results of the paper are clear and significant. The authors convincingly demonstrate that the waterfall features observed in ARPES experiments can be explained by the formation and splitting of a Hubbard band from the quasiparticle band within the single-band Hubbard model. This finding is important because it simplifies the understanding of the spectral anomalies by attributing them to local electronic correlations, as captured by the self-energy in DMFT. An interesting aspect of this work is the implicit use of DMFT's limitations as a strength. DMFT, by design, includes only local electronic correlations and does not account for non-local spin excitations in the self-energy. The authors capitalize on this limitation by showing that the waterfall phenomena can still be captured within this framework. This implies that the occurrence of waterfalls does not necessarily require the coupling of quasiparticles to non-local spin excitations, such as magnons. In a sense, the authors have turned a shortcoming of the DMFT approximation into a compelling feature of their analysis, reinforcing their conclusions.

I can wholeheartedly recommend the publication of this manuscript in Nature Communications due to its clear, simple yet broad key message, and its potential to make an important impact on a wide range of research in correlated electron systems, both experimental and theoretical. The authors present a compelling case that the waterfall features observed in ARPES spectra can be understood without invoking additional bosonic degrees of freedom. This insight is valuable and could influence future studies in the field.

However, I do have reservations regarding the use of the anthropomorphized metaphor of the "umbilical cord," and the title's reference to the "birth" of Hubbard bands. While such creative metaphors can be effective in conveying complex ideas, I find that this particular metaphor does not resonate well with the underlying physics. Specifically, the suggestion that Hubbard bands are "born" out of a non-interacting reference system seems artificial to me. In fact, one might argue the opposite: Hubbard bands are closely related to atomic-like excitations, while the low-energy quasiparticles are the emergent phenomenon when individual atoms are assembled into a solid. That said, this is a non-scientific critique, and I do not wish to dwell on it. The bottom line is that - in my opinion - the metaphor employed does not carry sufficient depth or accuracy to justify its use. I would suggest the authors reconsider the metaphor and perhaps choose a more fitting title that better reflects the scientific content and significance of their findings.

Let us start by thanking the Reviewer for the enthusiastic report and positive recommendation.

The only point raised is the Reviewer's concern regarding the metaphor "umbilical cord". We have received very positive feedbacks from colleagues during talks and discussions regarding this metaphor. Also when watching the enclosed movie, it really reminds us of an "umbilical cord". At least if the time direction refers to increasing the interaction strength, the Hubbard band develops and eventually splits up from the quasiparticle band. Hence in that sense the metaphor is quite fitting. As a matter of course a metaphor is never perfect. And, when taking reducing U as the time direction one can argue that the quasiparticle band develops. However, actually the same is true if you watch a movie of mother and child with time reversed. Let us also add that at finite doping and large interactions we always have both the quasiparticle and the Hubbard band, so our metaphor is indeed more fitting if one connects increasing U with the direction of time.

Following this discussion, we now explicitly relate the time direction with increasing U in the manuscript, but also emphasize that one could see it vice versa. We however would prefer to keep the metaphor. As a matter of course, if the Editor shares the same skepticism as the Reviewer and insists on its removal, we are open to making the change.